# Whole Transcriptome Sequencing Reveals Drought Resistance-Related Genes in Upland Cotton

**DOI:** 10.3390/genes13071159

**Published:** 2022-06-27

**Authors:** Juyun Zheng, Zeliang Zhang, Yajun Liang, Zhaolong Gong, Nala Zhang, Allah Ditta, Zhiwei Sang, Junduo Wang, Xueyuan Li

**Affiliations:** 1Cash Crops Research Institute of Xinjiang Academy of Agricultural Science (XAAS), Urumqi 830001, China; zjypp8866@126.com (J.Z.); zzldeyouxiang@126.com (Z.Z.); 13999966149@163.com (Y.L.); g15981775091@163.com (Z.G.); 13579975299@163.com (J.W.); 2Engineering Research Centre of Cotton, Ministry of Education, College of Agriculture, Xinjiang Agricultural University, 311 Nongda East Road, Urumqi 830052, China; znl1202022@163.com (N.Z.); s2564976727@163.com (Z.S.); 3Cotton Group, Plant Breeding and Genetics Division, Nuclear Institute for Agriculture and Biology (NIAB), Faisalabad 38000, Pakistan; adbotanist@yahoo.com; 4Nuclear Institute for Agriculture and Biology College (NIAB-C), Pakistan Institute of Engineering and Applied Sciences (PIEAS), Islamabad 45650, Pakistan

**Keywords:** upland cotton, drought stress, CeRNA, drought resistance gene, hydroponic

## Abstract

China, particularly the cotton-growing province of Xinjiang, is experiencing acute agricultural water shortages, stifling the expansion of the cotton sector. Discovering drought resistance genes in cotton and generating high-quality, drought-resistant cotton varieties through molecular breeding procedures are therefore critical to the cotton industry’s success. The drought-resistant cotton variety Xinluzhong No. 82 and the drought-sensitive cotton variety Kexin No. 1 were utilised in this study to uncover a batch of drought-resistant candidate genes using whole transcriptome sequencing. The following are the key research findings: A competing endogenous RNA network (*ceRNA*) was built using complete transcriptional sequencing to screen the core genes in the core pathway, and two drought-related candidate genes were discovered. It was found that γ-aminobutyric acid aminotransferase (*Gh**GABA-T*, *Gohir.A11G156000*) was upregulated at 0 h vs. 12 h and downregulated at 12 h vs. 24 h. L-Aspartate oxidase (*GhAO*, *Gohir.A07G220600*) was downregulated at 0 h vs. 12 h and upregulated at 12 h vs. 24 h. *GABA-T* is analogous to a pyridoxal phosphate-dependent transferase superfamily protein (*POP2*) in *Arabidopsis thaliana* and influences plant drought resistance by controlling γ-aminobutyric acid (*GABA*) concentration. The analogue of *GhAO* in *A. thaliana* is involved in the early steps of nicotinamide adenine dinucleotide (*NAD*) production as well as in plant antioxidant responses. This study revealed that gene expression regulatory networks can be used for rapid screening of reliable drought resistance genes and then utilised to validate gene function.

## 1. Introduction

Drought is a global issue that contributes to abiotic stress in plants throughout their lifecycle. The impacts of water scarcity on crops are complicated and diverse; they can delay plant growth and consequently change plant morphology and physiology [1,2]. Plants in drought-prone areas, however, have developed a variety of stress-resistance strategies, including developing larger and deeper root systems to increase water absorption from deep soil, regulating stomatal closure to reduce water loss, accumulating compatible solutes and protective proteins, and increasing antioxidant levels [3]. Reverse genetics aims to identify potential phenotypes that may result from a specific genetic sequence identified during DNA sequencing. Transgenic technology can be utilised to insert reverse resistance genes into plants to obtain novel materials as reverse resistance genetic engineering technology evolves. This not only solves the problem of cotton competing with food crops for water resources, but also raises farmers’ income, ensures cotton yield, and supports the textile industry’s continuing expansion.

There has been an increase in the number of studies on plant drought resistance mechanisms employing transcriptome sequencing in recent years, and a huge number of genes associated with drought resistance have been identified [4,5,6]. Transcriptome sequencing allows researchers to obtain transcripts for the expression of all genes in a single organism under precise physiological conditions or treatments, allowing them to study gene functions and structures on a global scale. Transcriptome sequencing has evolved into a valuable technique for investigating plant drought resistance mechanisms and identifying genes involved in drought resistance [7,8]. Bowman et al. [5] employed RNA-Seq to profile natural rain-fed and well-watered cotton in the field, and discovered 1530 divergent transcripts co-expressed in well-watered and natural rain-fed root tissues. Using transcriptome sequencing, Chen et al. [9] discovered that the upregulated genes in cotton under drought stress were primarily involved in glycerophospholipid metabolism; glycolysis/gluconeogenesis; amino and nucleotide sugar metabolism; lysosome, alanine, aspartate, and glutamate metabolism; fatty acid metabolism; pyruvate metabolism; and galactose metabolism pathways, such as cysteine and methionine metabolism. The downregulated genes were mostly involved in photosynthesis, photosynthetic antenna proteins, glyceride metabolism, oxidative phosphorylation, glycolysis/gluconeogenesis, phytohormone signalling, flavonoid biosynthesis, porphyrin and chlorophyll metabolism, and nitrogen metabolism. Through transcriptome sequencing, Zhang et al. [10] demonstrated that moderate liquorice drought could inhibit the expression of multiple enzymes in the cell wall and promote the synthesis pathways of terpenoids and flavonoids, revealing the molecular mechanism of liquorice adaptation to the accumulation of active components during drought. Transcriptome sequencing revealed that increased genes in rape roots under drought stress were primarily engaged in stimulated and stress-related biological processes, whereas upregulated genes in leaves were primarily involved in cells and cellular components [11]. Furthermore, the regulatory network of transcription factors under drought stress was investigated. Chen et al. [12] used RNA-Seq to examine the transcripts of soybean during drought circumstances, and the results revealed that dryness decreased photosynthesis and chlorophyll synthesis while promoting cell wall synthesis. According to Rohini et al. [13], the differentially expressed genes in chickpeas during drought stress were primarily involved in photosynthesis, trehalose synthesis, citrulline synthesis, UDP glucose (*UDP-Glc*) production, and other pathways.

In this study, whole transcriptome analysis was used to analyse Xinluzhong No. 82 and Kexin No. 1 under drought stress; ceRNA pathways were constructed and cotton drought-resistant pathways were screened. Our objectives were to obtain prospective drought-resistance genes, providing a theoretical framework for the isolation and identification of stress resistance-related genes in cotton and other crops.

## 2. Materials and Methods

### 2.1. Plant Materials and Samples

Drought-resistant Xinluzhong No. 82 and drought-sensitive Kexin No. 1 upland cotton (*G. hirsutum* L.) types were utilised in this experiment. The materials were provided by the Xinjiang Academy of Agricultural Sciences’ Upland Cotton Research Group of the Economic Crop Research Institute.

Seeds were germinated for three days on filter paper. Seedlings were transferred to hydroponic settings and cultivated for 25 days until the three-leaf one-heart stage. The water was changed every 4 days, and hydroponics were cultivated with 1/2 Hoagland nutrient solution. The treatment was only given to seedlings that grew consistently. The drought treatment with 17 percent Macrogol 6000 (*PEG 6000*) was utilised for both drought-resistant and drought-sensitive types at the three-leaf stage (24 days post-germination), and leaves were collected after treatment for 0 h, 12 h, and 24 h (see Figure 1).

### 2.2. RNA Extraction and Illumina Sequencing

Baierdi Biotechnology Co., Ltd. (Beijing, China) provided the Hoagland solution preparation, and TIANGEN Co., Ltd. provided the RNAprep Pure Polysaccharide Polyphenol Plant Total RNA Extraction Kit, FastKing RT Kit (with gDNase), and SuperReal PreMix Plus (SYBR Green) (Beijing, China).

Beijing Bomai Biotechnology Company produced the primers.

Each treatment was analysed in three duplicates, and total RNA was extracted using the RNAprep Pure Polysaccharide Polyphenol Plant Total RNA Extraction Kit (TIANGEN Co., Ltd., Beijing, China). UV absorption and denaturing agarose gel electrophoresis were used to assess the quality of the total RNA. The Illumina HiSeq high-throughput sequencing platform sequenced the cDNA library using sequencing by synthesis (SBS) and produced a vast volume of high-quality data (raw data).

### 2.3. Comparison with Reference Genome Sequences

For sequence alignment and subsequent analysis, the genome of *G. hirsutum* (UTX_TM1) was employed as a reference. Because of its great comparison efficiency, HISAT2 [14] is an efficient comparison system from the RNA sequencing experiment data.

### 2.4. Bioinformatics Analysis

After eliminating the spliced sequences and low-quality reads from the raw data, high-quality clean data were recovered; these clean data were utilised to perform expression analysis and new gene discovery. Advanced studies, such as the functional annotation and functional enrichment of differentially expressed mRNAs, miRNAs, lncRNAs, and circRNAs, were performed based on gene expression levels in various samples or groups.

Gene expression is temporal and place-specific, and both external and internal factors influence gene expression. Differentially expressed genes (DEGs) are those that have significantly varied expression levels under two different situations (control and treatment, wild-type and mutant, various time points, different tissues, etc.). Similarly, transcripts with significantly differing levels of expression are referred to as differentially expressed transcripts (DETs).

For all the samples in this investigation, there were two types of differential gene analysis: the comparison of differences within groups and the comparison of differences between groups. The comparison of intragroup differences refers to the treatment of the same material at different times, such as the comparison of the 16 percent PEG treatment with the 0 h treatment group at two different time points (12 h and 24 h); the comparison of differences between groups refers to the treatment of different materials at the same time, such as the comparisons 0hR vs. 0hS, 12hR vs. 12hS, and 24hR vs. 24hS (Appendix A).

The differentially expressed gene set is named using the “A vs. B” nomenclature and is the result of the differential expression analysis. Differentially expressed genes are classified as upregulated or downregulated based on the relative level of expression between two (groups) of samples. The expression levels of upregulated genes were higher in sample (group) B than in sample (group) A, while the inverse was true for downregulated genes. The order of A and B determines the relative upregulation and downregulation.

### 2.5. Analysis of Differential Gene Expression

As a measure of transcript and gene expression levels, FPKM [15] was employed to quantify lncRNA and gene expression. The TPM algorithm was used to normalise miRNA expression. The levels of circRNA expression in each sample were estimated, and junction reads were utilised to represent the levels of circRNA expression. The SRPBM [16] approach was used for standardisation.

### 2.6. Screening of Differentially Expressed Genes

DESeq2, as well as discrete convergence evaluation and multiple alterations, were used to improve the assessment’s stability and interpretability [17]. Fold change (FC) represents the ratio of expression between the two samples in the detection of differentially expressed genes (groups). FC > 1.5 and *p* < 0.05 indicated that the transcript or gene was differentially expressed between the two groups. The false discovery rate (FDR) is calculated by adjusting the difference’s *p*-value.

To collect annotation information for the target gene, BLAST was used to compare the projected target gene sequence with the NR, SWISS-PROT [18], GO [19], COG [20], KEGG [21], KOG [22], and Pfam [23] databases.

Differential genes were enriched, and clusterProfiler [24] used biological processes, molecular functions, and cell components in 2012. Enrichment analysis was performed using hypergeometric tests to identify GO and KEGG elements that were highly enriched when compared to the overall genetic background. We used the Database for Annotation, Visualization and Integrated Discovery (DAVID, 6.8) to perform GO enrichment analysis and KEGG pathway analysis on the differentially expressed mRNAs. The terms generated from the enrichment results were represented using a bar chart, a bubble plot, and so on.

### 2.7. qRT-PCR Analysis

In this study, one reference gene was selected as the candidate reference gene, and real-time RT- PCR primers were designed according to gene sequence. Primers that could amplify specific DNA fragments were screened for the subsequent stability analysis of candidate reference genes. To validate the accuracy of gene expression values derived from the transcriptome, nine DEGs with higher expression levels were chosen at random for qRT-PCR. The Primer3 website (Appendix A) produced primers based on the TIANGEN Co., Ltd. (Beijing, China) quick reverse transcription kit’s instructions, and they were amplified using SuperReal PreMix Plus (SYBR Green). The 2–ΔΔCt technique was used to calculate expression levels after normalising expression to the internal reference gene actin. In qRT-PCR, each sample was analysed using three biological and two technical replicates.

### 2.8. Statistical Analysis

All data generated from this experiment were analysed using the SPASS software. Data analysis was performed using SPSS22.0; *p* < 0.05 indicated that the difference was statistically significant. Quantitative data were presented as the mean ± SEM, which was used for comparison between groups by Student’s *t*-test, and the relative expression multiple differences were tabulated by 2–ΔΔCt method.

## 3. Results

### 3.1. Quality Control Results for the Total RNA

Following quality control, a total of 307.23 Gb of clean read data were received. Each sample had a Q30 base percentage of at least 93.73 percent and a GC content of at least 42.0 percent. The comparison efficiency between the reads of each sample and the reference genome ranged from 95.67 percent to 96.92 percent (Table 1), indicating that the data could be used for further investigation.

### 3.2. Sequencing Results for the Cotton Transcriptome Were Analysed

The four RNA types were predicted and identified using FASTQ data received from high-throughput sequencing (Appendix A). The entire amount of RNAs was counted, and known RNAs such as mRNA81336, lncRNA6906, miRNA453, and circRNA948 were included. The distribution of the sample points was shown on a two- or three-dimensional plane via the dimension reduction of the principal component analysis. Principal component analysis was used to cluster the samples using EIGENSOFT 7.2.1 software [25]. Samples treated for different periods were clearly separated, and the distance between biological duplicates demonstrated higher consistency between repeats and differences between different groups. The first three components explained 89.1 percent, 4.8 percent, and 1.9 percent of the variation, respectively. The biological replicates were assessed using Spearman’s correlation coefficient r (Spearman’s correlation coefficient). The closer r^2^ is to 1, the higher the correlation between the two repeated samples. In general, correlation between two repeated samples is regarded as being good when r2 is larger than 0.9. The results demonstrate that the correlation between the repeated samples of experimental materials was substantial, and that the difference between the two materials was significant, although the correlations between repeated samples were better (Figure 2).

### 3.3. Analysis of Differential Gene Expression

Many differentially expressed genes were discovered as a result of the sequencing data. The obtained *p* values were modified using Benjamini and Hochberg’s [26] false discovery rate (FDR). Significant differentially expressed genes were found using the following screening criteria: |log2(fold change)| > l, a fold change of 1.5, and a *p*-value of 0.01. There were 26,923 differentially expressed genes discovered based on gene expression levels in distinct samples. The majority of the comparisons found that there were more downregulated genes than upregulated ones. The number of differentially expressed genes from the same material processed at different times was far greater than the number of differentially expressed genes from different materials processed at the same time, and the statistics for the number of differentially expressed genes are shown in Table 2.

The fold change is the ratio of expression levels between two samples in the process of discovering differentially expressed lncRNAs (groups). A total of 6906 lncRNAs were discovered after high-throughput sequencing; 1639 were differentially expressed. As shown in the table below, the number of differentially expressed lncRNAs in each group was tallied (Table 3).

Hierarchical clustering analysis was performed on the identified differentially expressed lncRNAs, and lncRNAs with the same or comparable expression patterns were clustered. According to the heat map, most lncRNAs may be constitutively expressed under drought stress and induce a tolerance effect (Figure 3).

The screening criteria for discovering differentially expressed circRNAs were a fold change higher than or equal to 1.5 and a *p*-value of less than 0.05. The ratio of expression levels between two samples is represented by the fold change (groups). False positives will be a concern because circRNA differential expression analysis is an independent statistical hypothesis test on a large number of circRNA expression levels [27,28]. As a result, throughout the analytical procedure, the *p*-value is used as a critical indicator for screening differentially expressed circRNAs. The prediction engine detected a total of 948 circRNAs, with a modest number of differentially expressed. The statistics for the number of differentially expressed circRNAs in each group are shown in Table 4.

The screened differentially expressed circRNAs were subjected to a hierarchical clustering analysis. XJ0hR was found to have more differences in differentially expressed circRNAs than XJ0hS (Figure 4).

The screening criteria for discovering differentially expressed miRNAs were |log2(FC)| ≥ 0.58 and a *p*-value of ≤0.05. The ratio of expression levels between two samples is represented by the fold change (FC) (groups). The original hypothesis’ significance *p*-value can be stated as the probability of no difference in expression. False positives will be a concern since the differential expression analysis of miRNAs is an independent statistical hypothesis test for a large number of miRNA expression levels. As a result, the Benjamini–Hochberg adjustment method is commonly utilised in the analysis process to determine whether the initial hypothesis is significant. The *p*-value was adjusted, and the false discovery rate (FDR) was finally utilised as the primary indication for filtering differentially expressed miRNAs. There were 453 miRNAs found in total, comprising 60 known miRNAs and 393 newly predicted miRNAs. The number of miRNAs that were differently expressed was modest (Table 5).

Hierarchical clustering analysis was performed on the identified differentially expressed miRNAs, and miRNAs with the same or comparable expression behaviour were clustered. XJ12hS2 had many more differentially expressed miRNAs than XJ24hR1, but XJ24hR1 had significantly fewer. The clustering outcomes are as shown in Figure 5.

Nine genes were chosen at random from the significantly expressed genes for verification using real-time fluorescence quantitative PCR. The difference in expression between upregulated and downregulated genes was consistent with the sequencing data’s trend, showing that the sequencing results were trustworthy (Figure 6).

### 3.4. Analysis of Differentially Expressed Genes’ Functional Annotation and Enrichment

The table below shows the statistics for the number of genes annotated in each differentially expressed gene collection (Table 6).

ClusterProfiler [24] was used to look for genes that were enriched in biological processes, molecular functions, and cell component categories. Enrichment analysis employs a hypergeometric test approach to identify GO and KEGG elements that are considerably enriched when compared to the overall genetic background. GO entries were considerably enriched in heterocyclic chemical binding, with reactions to redox status and pigment binding being the most enriched activities. The grouping of pathways indicates the significance of specific biological activities in an indirect way. Signalling pathways such as plant hormone signal transduction were considerably enriched in KEGG entries. Thiamine metabolism was the most strongly expressed signalling pathway in the 12 h treatment of the two types. Thiamine metabolism is essential for plant growth and development, as well as biological and non-biological stress responses. The glyoxylate and dicarboxylate metabolism signalling pathway was strongly elevated in all treatments (Figure 7), which is consistent with the intricacy of the drought stress response.

GO and KEGG enrichment analyses were performed on differently expressed lncRNAs, and differentially expressed lncRNA target genes were clustered as well. Heterocyclic compound binding was the term that was substantially enriched with differentially expressed genes, while organelle membrane and changed amino acid binding were the most enriched functions (Appendix A). Photosynthesis, ascorbate and aldarate metabolism, glyoxylate and dicarboxylate metabolism, and inositol phosphate metabolism were found to be the signalling pathways with the most enriched genes (Figure 8). The following terminology emerged from a comprehensive review of the positive and negative impacts of LTGs under drought stress: microtubule-based process (50), organelle membrane (2), nitrogen metabolism (16), cyanoamino acid metabolism (1), and inositol phosphate metabolism (1), (8). These pathways are critical regulators in the drought response process.

We performed functional annotations on differentially expressed circRNAs, and the annotation results are presented in the table below (Table 7).

Biological processes, molecular functions, and cellular components are the three primary categories of GO. The most relevant BP (biological process) terms, according to the analysis, were responses to stimuli and single-organism processes, and the most relevant CC (cellular component) phrases were the cell, cell part, membrane, membrane part, and organelle. The word with the highest MF (molecular function) enrichment was associated with binding and catalytic activity (Appendix A).

The statistical results of the number of differentially expressed miRNA target genes annotated between samples are shown in the table below (Table 8).

GO enrichment analysis revealed that in the BP category, metabolic processes, cellular processes, and single-organism processes were highly enriched; in the CC category, the membrane, cell, cell part, membrane part, and organelle were highly enriched; and in the MF category, binding and catalytic activities were highly enriched (Appendix A).

The KEGG pathway enrichment analysis of the differentially expressed miRNA target genes was performed to determine whether the differentially expressed miRNA target genes were overrepresented in a certain pathway. The enrichment factor was used to analyse the enrichment degree of the pathway, and Fisher’s exact test method was used to calculate the significance of enrichment. The main enrichment pathways were as follows: galactose metabolism; flavonoid biosynthesis; N-glycan biosynthesis; alanine, aspartate and glutamate metabolism; pentose and glucuronate interconversions; SNARE interactions in vesicular transport; oxidative phosphorylation; TCA cycle; circadian plant; and the mRNA surveillance pathway (Appendix A).

### 3.5. Construction of the ceRNA Regulatory Network and Integration Analysis of Key Pathways

*CeRNA* is a novel transcriptional regulatory molecule. *CeRNAs* can bind to the same miRNA in a competitive manner via microRNA response elements (MREs) to regulate their expression levels [29]. The ceRNA hypothesis elucidates a novel *RNA* binding mechanism. CeRNAs can regulate the expression of other ceRNAs by competitively binding to the miRNA. For example, the microRNA can cause gene silencing, and the lncRNA competitively binds to the microRNA, affecting the function of the microRNA leading to gene silencing. Here, mRNA, lncRNA, and circRNA can be used as ceRNAs. We employed miRNA targeting to find possible ceRNA interactions in this investigation. Cytoscape was used to create a ceRNA connection network, which had 1289 edges and 1039 points, including 146 lncRNAs, 859 mRNAs, and 34 circRNAs (Figure 9).

Each group of differently expressed RNAs extracted from the ceRNA relationship pair brings us one step closer to the network, and then the differential ceRNA relationship pair can be obtained (Appendix A). The standard algorithm PageRank in random walk is used to obtain the scores (that is, the importance) of all nodes in the network (that is, the different ceRNAs). The important RNA was chosen as the key research object if it was ranked in the top 0.05 points in the network. Pathway enrichment analysis was performed on genes in key nodes + lncRNA target genes + circRNA host genes (hence, they are referred to as “key genes”), and the top five pathways with the highest enrichment were chosen. The relationships between all genes and genes in these five pathways were retrieved and integrated into a pathway network, and essential genes were assigned to pathways. The node can be thought of as the ceRNA’s (gene/lncRNA/circRNA) ID in this case.

The top pathways in terms of gene enrichment are ribosome biogenesis in eukaryotes; circadian rhythm in plants; N-glycan biosynthesis; the pentose phosphate pathway; alanine, aspartate, and glutamate metabolism; arginine biosynthesis; folate biosynthesis; 2-oxocarboxylic acid metabolism; glyoxylate and dicarboxylate metabolism; and other glycan degradation (Figure 10).

### 3.6. Differential Gene Annotation and ceRNA Network Analysis Were Used to Screen Drought Resistance-Related Genes

It was discovered that there may be drought-related genes that are important genes in the alanine, aspartate, and glutamate metabolism pathway (ko00250) by the integration analysis of key genes in the key pathways and the annotation analysis of GO and KEGG, together with past research advancements (Table 9). For the two key genes in this pathway (Appendix A). The *Gohir.A11G156000* (Appendix A) and *Gohir.A07G220600* (Appendix A) genes were annotated and tested. In the same variety, compared to the 0 h treatment, the *Gohir.A11G156000* was upregulated and the *Gohir.A07G220600* was downregulated in the 12 h treatment. The *Gohir.A11G156000* gene was downregulated in the 24 h therapy compared to the 12 h treatment, while the *Gohir.A07G220600* gene was increased. The functions of these two genes were discovered using GO and KEGG analyses. The co-expression trend analysis of the identified drought resistance genes offered more intuitive knowledge of these genes’ expression trends across samples. *Gohir.A11G156000* was elevated at 0 h compared to 12 h and downregulated at 12 h compared to 24 h. *Gohir.A07G220600* was downregulated at 0 h compared to 12 h and elevated at 12 h compared to 24 h. The comparison of *Gohir.A11G156000* with Tair3 shows that the homologous gene in A. thaliana is *AT3G22200*, with 77% nucleotide homology and 77% protein homology, and the gene is *POP2*, a pyridoxal phosphate (PLP)-dependent transferase superfamily protein. The homologous gene for *Gohir.A07G220600* in *A. thaliana* is *AT5G14760*, with 76% nucleotide homology and 79% protein homology, which encodes an *AO-L-aspartate oxidase* (Figure 11).

## 4. Discussion

### 4.1. Rapid Screening of Drought Resistance-Related Genes by Differential Gene Annotation and ceRNA Regulatory Network Analysis

*CeRNA* can be employed as a bait to attract and isolate a miRNA, thereby relieving the miRNA’s repression of target genes. Using MRE as the communication language, mRNA, lncRNA, circRNA, and pseudogenes can achieve mutual regulation via miRNA competition processes, resulting in the formation of a large-scale posttranscriptional regulatory network known as a *ceRNA* network.

In Arabidopsis, Franco-Zorrilla et al. [30] discovered the first noncoding RNAs with miRNA bait functions. Under low phosphorus conditions, lncRNA IPS1 competed with PH02 for the binding of miR399. Under salt stress, the whole transcriptome sequencing [31] of cotton trifoliate seedlings revealed 44 lncRNAs and 6 miRNAs. qRT-PCR was used to confirm the differential expression, and lnc973 and lnc253 competed to bind miR399 and miR156e, respectively.

Despite the fact that *ceRNA* was first discovered in plants, corresponding research in humans and mammals has very much lagged behind. From 2011 to 2018, the number of ceRNA-related research studies included in PubMed increased, although the number of plant-related *ceRNA* reports remained limited. In 2018, the proportion of plant-related reporting in 688 articles was only 9.74%.

The retrieved genes were coupled with the cotton genome database in this investigation via the functional annotation of differential genes and ceRNA network screening, and finally, the key genes were determined. This is a quick way to obtain drought-related genes. More key regulatory genes that can directly act on and assist plants’ response to drought stress signals were identified through the creation of a ceRNA network. Finally, two drought-related candidate genes were discovered to have elevated expression during drought stress.

### 4.2. Study of GABA in Plant Stress Resistance

The *POP2* gene, which encodes *GABA-T*, was predicted to be the *Gohir.A07G220600* homologous gene (GABA aminotransferase). *GABA-T* catalyses the conversion of *GABA* in the *GABA* shunt pathway to *SSA* (hemisuccinic acid), and it has been linked to plant stress resistance in some studies. Together, glutamic acid decarboxylase (*GAD*) in the cytoplasm and *GABA* transaminase (*GABA-T*) and succinic semialdehyde dehydrogenase (*SSADH*) in the mitochondria regulate *GABA* branch metabolism in plants, with *GAD* being the rate-limiting enzyme for *GABA* synthesis. *GABA* is a nonprotein amino acid with four carbons and the chemical formula C4H9NO2 (*GABA*). It has sparked widespread interest since it was discovered in potato tubers in the 1950s. *GABA* has been discovered to be abundant in vertebrates, plants, and microbes in subsequent investigations. *GABA* is synthesised in the cytoplasm [32]. The *GABA* concentration in plant tissues is quite low, often ranging between 0.3 and 32.5 mol/g. It has been reported that *GABA* accumulation in plants is related to stress response, which can lead to the rapid accumulation of *GABA* under hypoxia, heat, cold, mechanical injury, and salt stress. *GABA* metabolism in higher plants is mostly completed by three enzymes. First, *GAD* causes irreversible decarboxylation of L-glutamic acid at the γ-position, and the released products *GABA*, pyridoxal 5-phosphate monohydrate (*PLP*), and *GAD* return to their starting states at the end of the reaction. *GABA* then combines with pyruvate and α-ketoglutaric acid to generate succinic semialdehyde under the catalysis of *GABA-T*. Finally, *SSADH* catalyses the oxidative dehydrogenation of succinic semialdehyde to generate succinic acid, which is then recycled through the tricarboxylic acid cycle (Krebs cycle) [31,33]. This metabolic route contributes to the *GABA* shunt, a branch of the tricarboxylic acid (TCA) cycle. Previous research has showed that changes in the TCA cycle’s enzyme activity indicated *GABA’s* metabolic role and the strong link between *GABA* and respiration [34,35]. *GABA* is also an oxidative metabolite regulator. When the *A. thaliana SSADH* mutant was subjected to high temperatures, reactive oxygen intermediates (ROIs) accumulated, causing plant mortality, demonstrating a link between ROIs and *GABA*. Similarly, at high temperatures, *SSADH* and *GABA-T* gene mutants exhibit a large number of *ROIs*. The administration of the ROI removal drug PBN can promote *GABA* accumulation, increasing yeast survival. As a result, it is thought that the *GABA* shunt process can minimise the formation of *ROIs*, thus protecting organisms from oxidative damage and decay induced by stress.

The following conclusions can be drawn from the preceding analysis. The *Gohir.A07G220600* gene was increased in the alanine, aspartate, and metabolic pathways at 0h versus 12h, demonstrating that *GABA-T* accumulated rapidly in cotton under dry conditions. *GABA* has the potential to lessen the damage caused by reactive oxygen species to stressed plants while also increasing the activity of defensive enzymes in plants. The difference in downregulation between 12 and 24 h revealed that *PLP* and *GAD* recovered to their starting states at the end of the reaction. Drought decreased the growth and leaf area extension of roots and stems, while reactive oxygen species levels increased. The production of low molecular osmotic adjustment chemicals, such as *GABA* and other amino acids, polyols, and organic acids increased, as did the expression of antioxidant damage enzymes [36].

### 4.3. Mechanism of L-Aspartate Oxidase and NAD in Plant Stress Resistance

The findings of the homology comparison within *A. thaliana* predicted that the homologous sequence of *AO* was *Gohir.A07G220600*, which is a L-aspartate oxidase encoding the early phases of *NAD* production [37]. *NAD* is a ubiquitous coenzyme that participates in redox reactions and can be converted between oxidised and reduced forms with no net expenditure. In a catabolic reaction, *NAD* is primarily transformed into its reduced form *NADH*, and *NADH* is primarily oxidised via the mitochondrial electron transfer chain and participates in the redox process of cells. *NADH* can reduce peroxidation by reacting with free radicals. Arabidopsis generates *NAD* from Asp by utilising *AO*, *QS*, and QPT, all of which are required for plant growth and development. These enzymes are discovered in the plastid, implying that the first stages in *NAD* production take place in this organelle [38]. The reaction of Asp catalysed by L-Asp oxidase has the following equation: L-aspartic acid + O_2_ = iminoaspartic acid (iminosuccinic acid) + H_2_O_2_. The findings revealed that L-Asp oxidase was linked to an increase in reactive oxygen species [39].

Based on the preceding investigation, we can draw the following conclusions: The *Gohir.A07G220600* gene was downregulated in the alanine, aspartate, and glutamate metabolic pathways in 0 h vs. 12 h, indicating cotton stomatal closure during drought conditions, which inhibited respiration and lowered the formation of reactive oxygen species to increase stress resistance. The 12 h vs. 24 h upregulation demonstrated that under drought stress, reactive oxygen species in plants rose dramatically. L-Aspartate oxidase catalysed the binding of L-aspartate to oxygen, resulting in a ROS burst. The enhanced expression of L-aspartate oxidase increased *NADH* production, which improved the plant’s antioxidant capability. Drought decreased the growth and leaf area extension of roots and stems, while reactive oxygen species increased. The production of low-molecular-weight osmotic adjustment chemicals increased, such as *GABA* and other amino acids, polyols, and organic acids, as did the expression of antioxidant damage enzymes.

Drought increases the expression of genes involved in intracellular homeostasis, active oxygen scavenging, structural protein stability protection, osmotic regulators, transporters, and other processes. Highly expressed genes were engaged in controlling a range of drought resistance-related pathways in this experiment, and they cooperated to improve plant drought resistance. Simultaneously, the findings verified the efficacy of the expression regulation network, which might be employed to investigate more multifunctional genes.

## 5. Conclusions

This is the first study to use three-leaf cotton plants in hydroponic studies with a 17% PEG solution to simulate drought stress. We examined the transcription levels of all genes in leaf samples from different types and treatment timepoints at 0 h, 12 h, and 24 h. We were able to screen for drought resistance-related pathways in cotton, key genes in KEGG pathways, and candidate salt-resistance genes using the GO clustering of differentially expressed genes, KEGG clustering, pathway analysis, and the construction of ceRNA networks, as well as the synthesis of existing literature reports. The primary conclusions are described below.

The KEGG analysis of the differentially expressed genes reveals enrichment in metabolic pathways such as plant hormone signal transduction, carbon metabolism, and photosynthesis-related pathways, implying that these processes may be important in plant growth and development as well as in drought resistance. Hormones in the phytohormonal signalling pathway interact with one another and can act as secondary signalling molecules to control the expression of downstream stress-related genes. Carbohydrates can be employed not only as an energy source, but also as signalling molecules to govern plant development and resistance to adversity. Photosynthesis-related pathways have also been found to be considerably enriched. Organic matter created by photosynthesis can provide energy for plant growth and development. Cotton has the ability to induce a large number of functional proteins that enable it adapt to harsh circumstances and boost its resistance to stress. Cotton inhibits oxidative damage under drought stress by generating a large number of enzymes involved in ROS elimination.

Two drought-related candidate genes were identified (*Gohir.A11G156000* and *Gohir.A07G220600*). Following genetic function verification, the dependability of drought-resistant genes can be swiftly screened utilising ceRNA regulatory networks.

## Figures and Tables

**Figure 1 genes-13-01159-f001:**
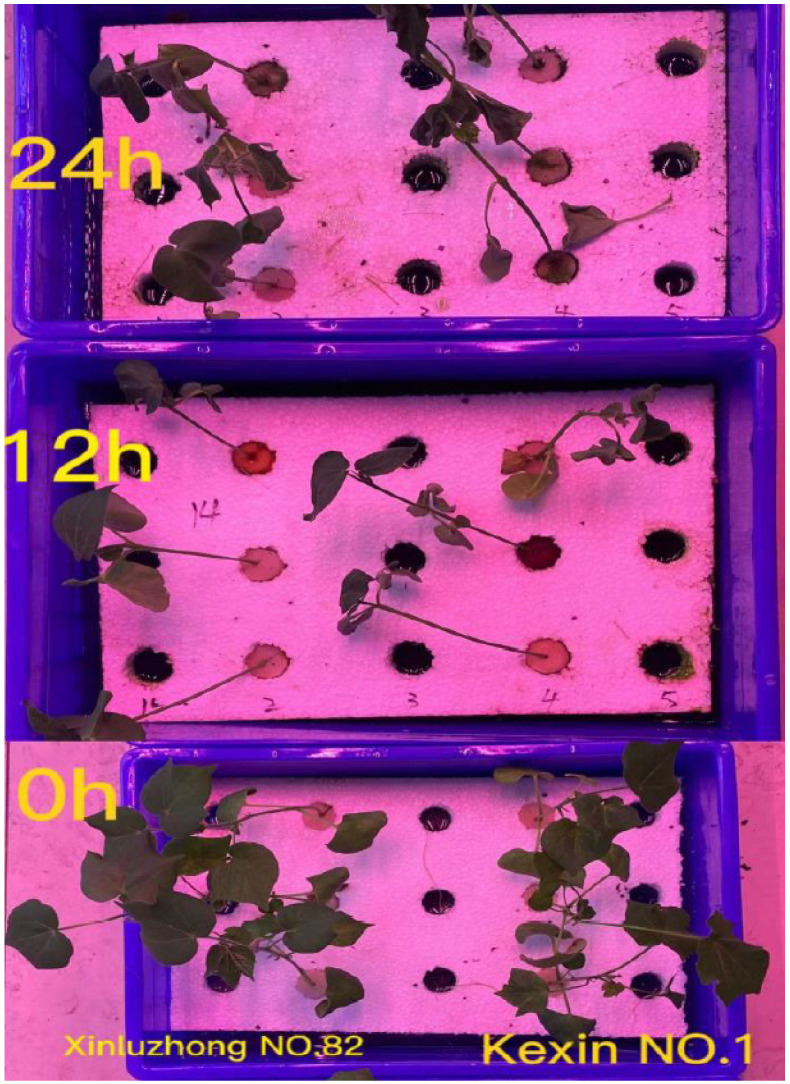
The performance of Xinluzhong No. 82 and Kexin No. 1 under seedling drought stress.

**Figure 2 genes-13-01159-f002:**
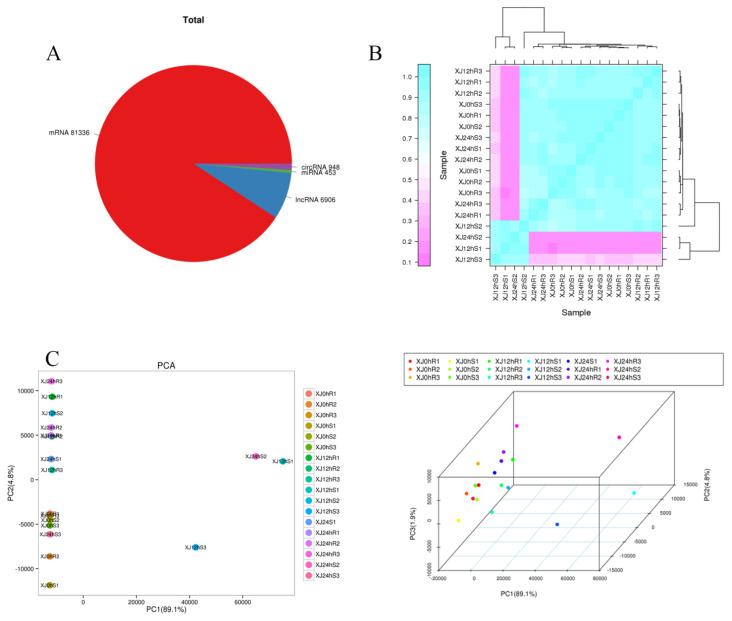
(**A**) Statistics for amounts of various RNA; (**B**) sample correlation diagram, with different colours representing different correlation coefficient values. The abscissa and ordinate indicate separate samples. (**C**) A two-dimensional PCA clustering map and a three-dimensional PCA clustering map. The samples are gathered into three dimensions by PCA: PC1 represents the first principal component, PC2 represents the second principal component, and PC3 represents the third principal component, a point represents a sample, and a colour represents a grouping.

**Figure 3 genes-13-01159-f003:**
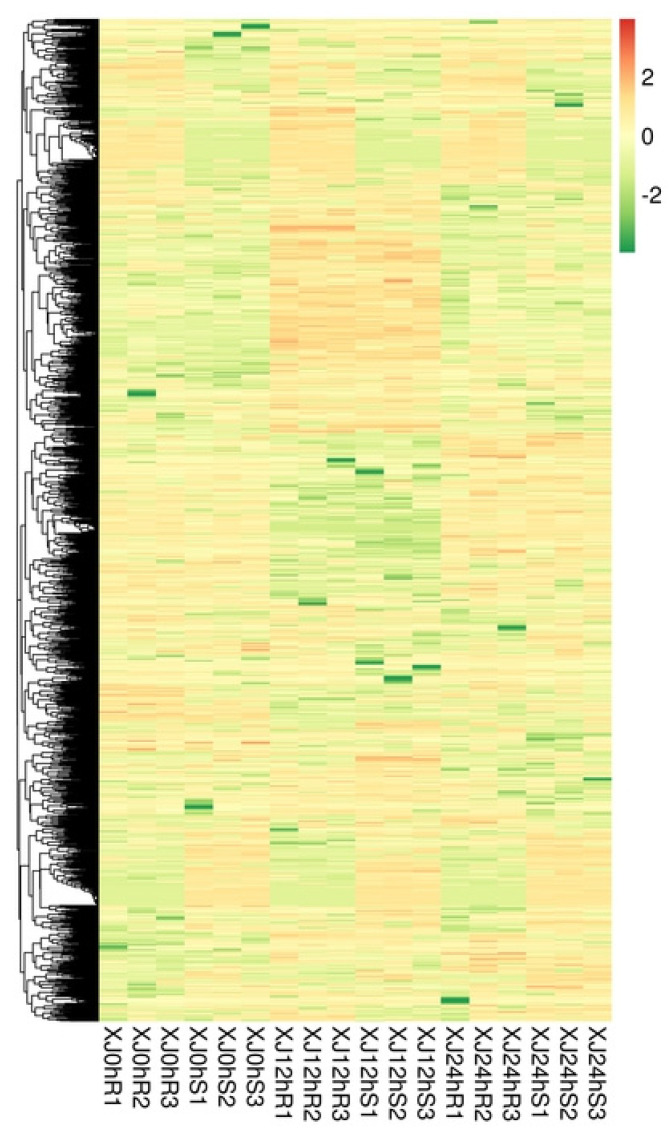
Differentially expressed lncRNA cluster map. Note: The figure’s columns indicate distinct samples, whereas the rows show different lncRNAs. The colour denotes the expression level of lncRNAs in the sample as log10 (FPKM + 0.000001).

**Figure 4 genes-13-01159-f004:**
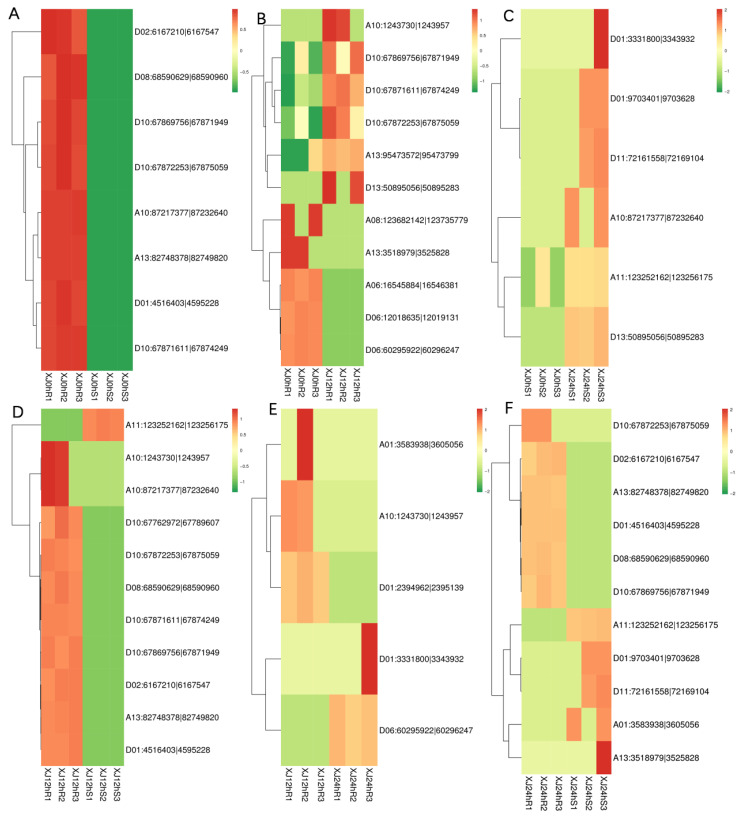
Cluster map of differentially expressed circRNA expression patterns. Note: Different columns in the figure represent different samples, and different rows represent different circRNAs. The colour represents the expression level of the circRNA in the sample (log10SRPBM+0.000001). (**A**) XJ0hR1_XJ0hR2_XJ0hR3_vs_XJ0hS1_XJ0hS2_XJ0hS3; (**B**) XJ0hR1_XJ0hR2_XJ0hR3_vs_XJ12hR1_XJ12hR2_XJ12hR3; (**C**) XJ0hS1_XJ0hS2_XJ0hS3_vs_XJ24hS1_XJ24hS2_XJ24hS3; (**D**) XJ12hR1_XJ12hR2_XJ12hR3_vs_XJ12hS1_XJ12hS2_XJ12hS3; (**E**) XJ12hR1_XJ12hR2_XJ12hR3_vs_XJ24hR1_XJ24hR2_XJ24hR3; (**F**) XJ24hR1_XJ24hR2_XJ24hR3_vs_XJ24hS1_XJ24hS2_XJ24hS3.

**Figure 5 genes-13-01159-f005:**
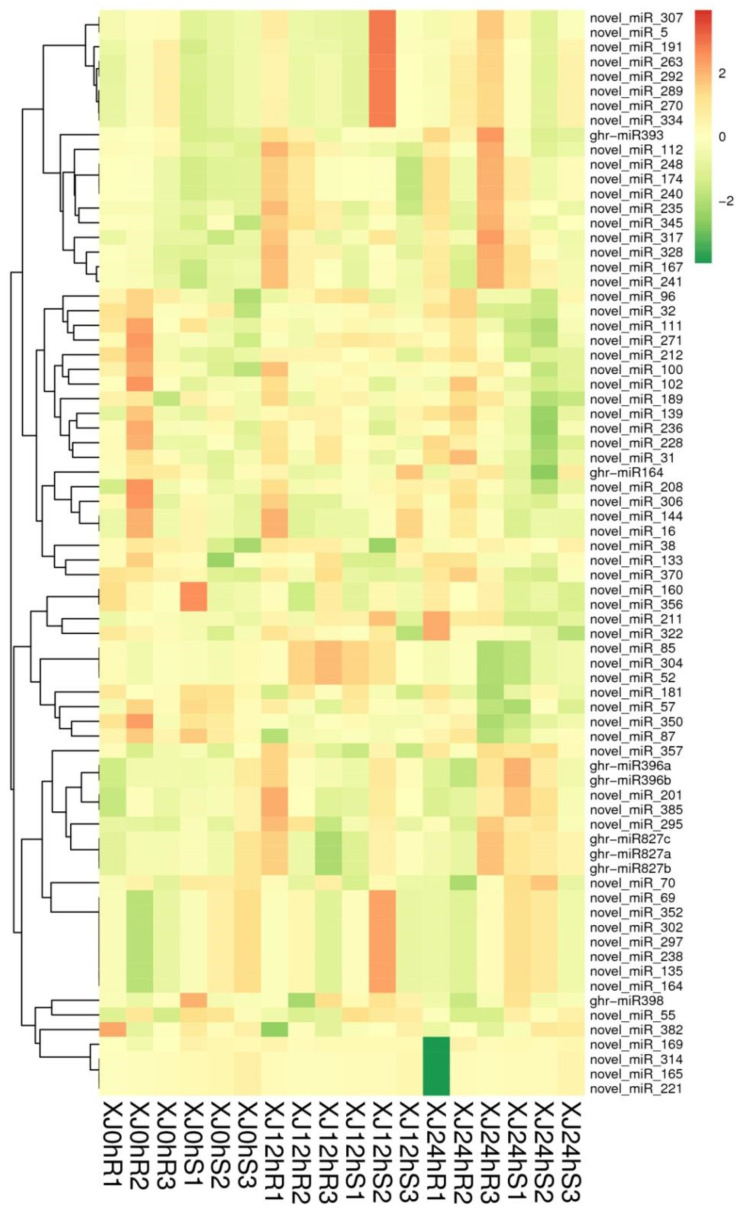
Cluster map of differentially expressed miRNAs. Note: The diagram above depicts a clustering map of differentially expressed miRNAs. The columns indicate various samples, while the rows represent different miRNAs. Log10 (TPM + 1 × 10^−6^) values are used for clustering. MiRNAs with high expression are represented by red, while miRNAs with low expression are represented by green.

**Figure 6 genes-13-01159-f006:**
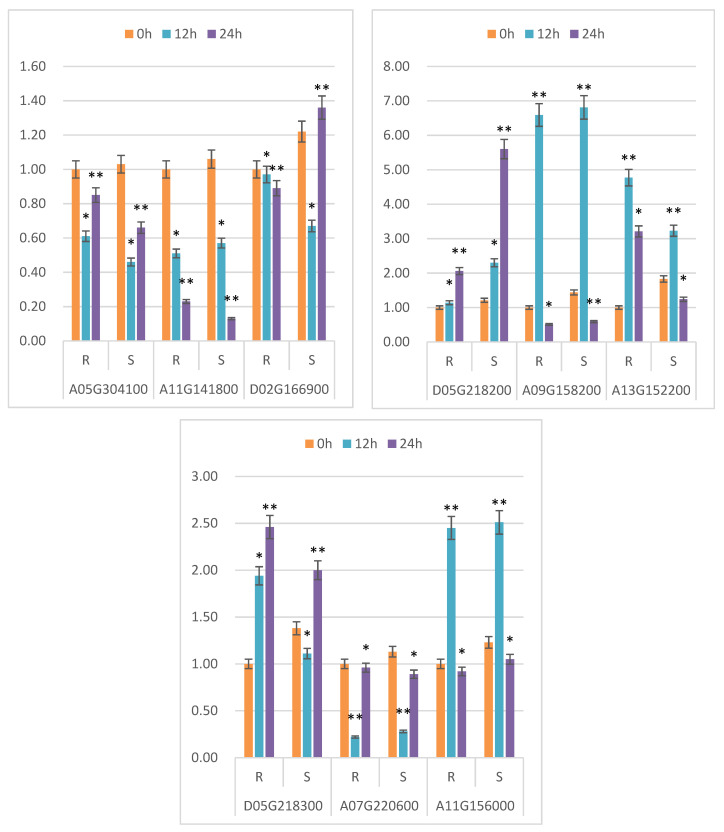
qRT-PCR validation of differentially expressed genes. Note: 0 h, 12 h and 24 h indicate different treatment times; R represents the drought-resistant material; S represents sensitive material; abscissa represents different genes; and ordinate represents the relative expression of genes between different materials. Transcriptome expression: under the accession number PRJNA769509, which is publicly accessible at https://www.ncbi.nlm.nih.gov/ (accessed on 1 January 2021) * and ** indicate significant differences at the 0.05 and 0.01 level, respectively.

**Figure 7 genes-13-01159-f007:**
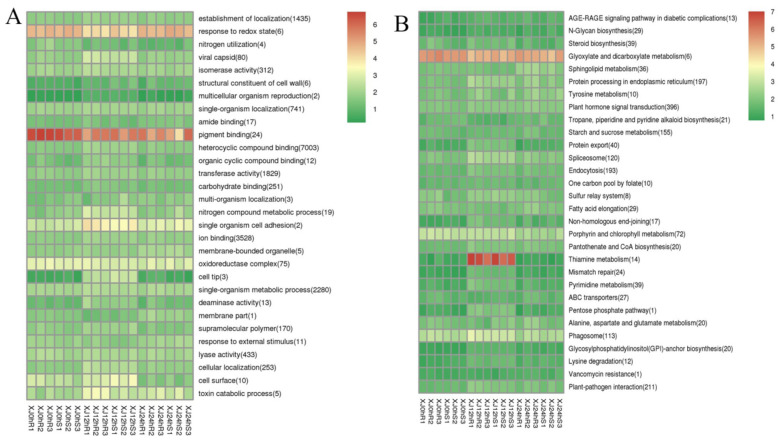
(**A**): GO enrichment cluster map for differentially expressed genes; (**B**): KEGG enrichment cluster map for differentially expressed genes. Note: Red indicates metabolic pathways with high expression, and blue indicates metabolic pathways with relatively low expression. The parentheses after the label for each metabolic pathway include the number of genes in the metabolic pathway with significant differences.

**Figure 8 genes-13-01159-f008:**
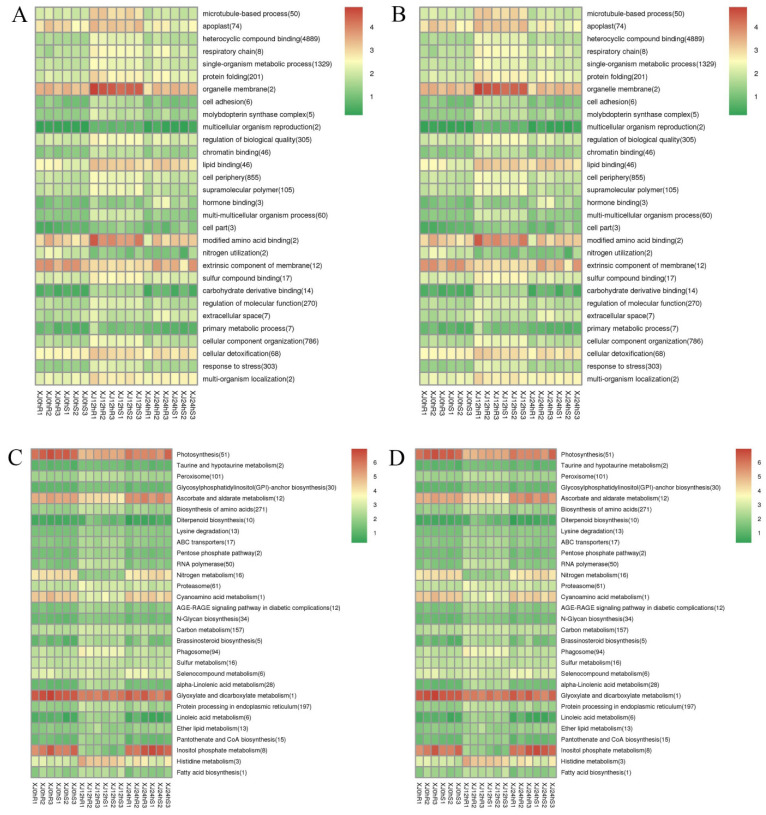
(**A**) Clustering graph of the GO enrichment of differentially expressed lncRNA cis-target genes. (**B**) Clustering graph of the GO enrichment of differentially expressed lncRNA trans-target genes. (**C**) Cluster map of the KEGG enrichment of differentially expressed lncRNA cis-target genes (**D**) Cluster map of the KEGG enrichment of differentially expressed lncRNA trans-target genes. Note: Red indicates metabolic pathways with high expression, and green indicates metabolic pathways with relatively low expression. The parentheses after the label of each metabolic pathway include the number of genes with significant differences in the metabolic pathway.

**Figure 9 genes-13-01159-f009:**
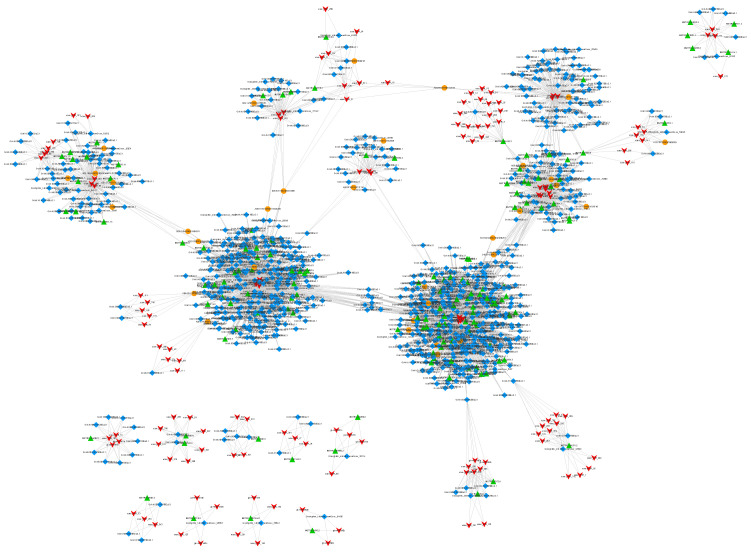
ceRNA regulation network. Note: circRNA: orange circles. Gene: blue diamonds. lncRNA: green triangles. miRNA: red arrows.

**Figure 10 genes-13-01159-f010:**
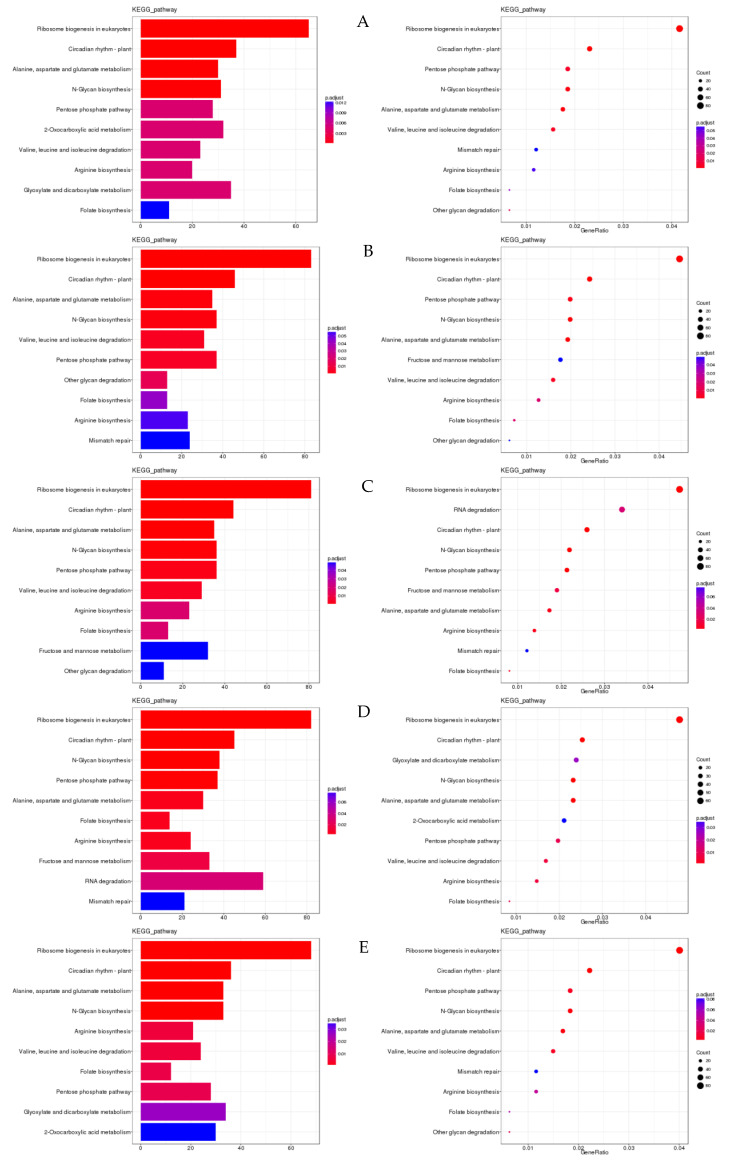
Bar chart and bubble chart of KEGG enrichment. Note: KEGG enrichment bar graph: the abscissa is GeneNum, which is the number of genes of interest annotated in the entry, and the ordinate is each pathway entry. The colour of the column represents the *p*-value of the hypergeometric test. KEGG enrichment bubble chart: the abscissa is GeneRatio, which is the ratio of the gene of interest annotated in this entry to the number of differentially expressed genes, and the ordinate is each pathway entry. The size of the dot represents the number of differentially expressed genes annotated in the pathway, and the colour of the dot represents the *p*-value of the hypergeometric test. (**A**) XJ0hR1_XJ0hR2_XJ0hR3_vs_XJ0hS1_XJ0hS2_XJ0hS3; (**B**) XJ0hR1_XJ0hR2_XJ0hR3_vs_XJ12hR1_XJ12hR2_XJ12hR3; (**C**) XJ0hR1_XJ0hR2_XJ0hR3_vs_XJ24hR1_XJ24hR2_XJ24hR3; (**D**) XJ0hS1_XJ0hS2_XJ0hS3_vs_XJ12hS1_XJ12hS2_XJ12hS3; (**E**) XJ0hS1_XJ0hS2_XJ0hS3_vs_XJ24hS1_XJ24hS2_XJ24hS3; (**F**) XJ12hR1_XJ12hR2_XJ12hR3_vs_XJ12hS1_XJ12hS2_XJ12hS3; (**G**) XJ12hR1_XJ12hR2_XJ12hR3_vs_XJ24hR1_XJ24hR2_XJ24hR3; (**H**) XJ12hS1_XJ12hS2_XJ12hS3_vs_XJ24hS1_XJ24hS2_XJ24hS3; (**I**) XJ24hR1_XJ24hR2_XJ24hR3_vs_XJ12hS1_XJ12hS2_XJ12hS3.

**Figure 11 genes-13-01159-f011:**
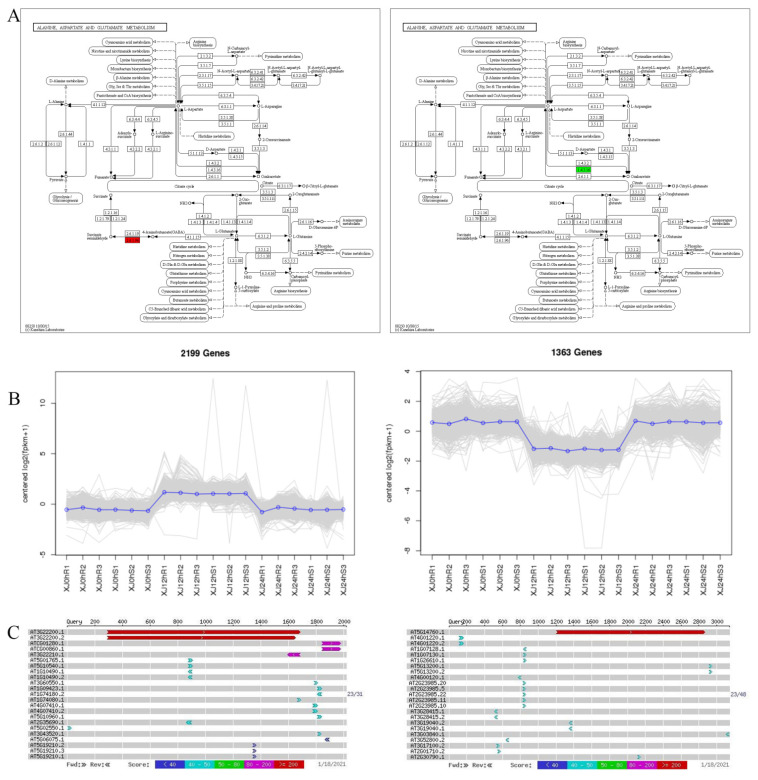
Drought-resistant gene screening. Note: (**A**) KEGG annotation path diagram. The number in the box is the EC number of the enzymes; the entire pathway is composed of a complex biochemical reaction catalysed by a variety of enzymes. In this pathway diagram, the enzymes related to differentially expressed genes are marked with different colours. (**B**) Gene co-expression trend graph. The abscissa represents the sample or time point, and the ordinate represents logarithmic centred expression levels. The grey line in the figure represents the expression trend of a gene. The blue line indicates the type of gene. (**C**) Gene sequence BLAST results.

**Table 1 genes-13-01159-t001:** Statistical table for the sequencing data evaluation.

SampleID	ReadSum	BaseSum	GC (%)	N (%)	Q20 (%)	Q30 (%)	Total Reads	Mapped Reads
XJ0hR1	71,742,474	21,224,569,062	42.85	0	97.92	93.80	143,484,948	138,314,357 (96.40%)
XJ0hR2	53,952,826	16,058,270,844	42.78	0	97.93	93.88	107,905,652	104,104,314 (96.48%)
XJ0hR3	67,100,443	19,944,130,312	43.07	0	97.90	93.73	134,200,886	129,284,996 (96.34%)
XJ0hS1	53,931,117	16,041,131,562	42.77	0	98.34	94.70	107,862,234	103,981,706 (96.40%)
XJ0hS2	57,233,313	17,014,361,012	42.72	0	98.32	94.65	114,466,626	110,557,335 (96.58%)
XJ0hS3	56,289,536	16,740,674,196	42.84	0	98.21	94.50	112,579,072	108,590,412 (96.46%)
XJ12hR1	60,305,370	17,970,064,928	43.00	0	98.27	94.52	120,610,740	116,216,676 (96.36%)
XJ12hR2	53,171,614	15,800,217,442	42.67	0	98.28	94.59	106,343,228	102,417,327 (96.31%)
XJ12hR3	54,393,796	16,198,260,692	42.06	0	98.29	94.60	108,787,592	104,988,226 (96.51%)
XJ12hS1	55,632,640	16,553,339,614	42.5	0	98.27	94.51	111,265,280	107,405,318 (96.53%)
XJ12hS2	55,334,885	16,503,777,878	42.61	0	98.19	94.39	110,669,770	105,879,284 (95.67%)
XJ12hS3	56,241,086	16,733,164,858	42.56	0	98.20	94.39	112,482,172	107,971,513 (95.99%)
XJ24hR1	55,174,614	16,453,704,450	42.76	0	98.21	94.48	110,349,228	106,142,505 (96.19%)
XJ24hR2	55,843,015	16,660,958,134	42.43	0	98.34	94.76	111,686,030	107,264,860 (96.04%)
XJ24hR3	58,592,920	17,464,204,712	42.98	0	98.22	94.44	117,185,840	113,571,833 (96.92%)
XJ24hS1	55,363,695	16,530,287,984	42.26	0	98.47	94.98	110,727,390	106,999,856 (96.63%)
XJ24hS2	55,231,921	16,459,828,202	42.49	0	98.19	94.32	110,463,842	106,313,930 (96.24%)
XJ24hS3	56,920,057	16,913,326,528	42.8	0	98.45	94.91	113,840,114	109,889,372 (96.53%)

Note: SampleID: sample name. ReadSum: the total amount of clean paired-end reads. BaseSum: The sum of the clean data’s base numbers. GC (percent): the percentage of G and C bases in the total bases in the clean data. N (percentage): the percentage of unresolved bases in the clean data. Q30 (percentage): the proportion of bases with clean data mass values greater than or equal to Q30. Total Reads: the number of single-ended clean reads. Mapped Reads: the number of reads that have been mapped to the reference genome, as well as the proportion of clean reads. XJ represents Xinjiang; 0h, 12h, and 24h indicate different treatment times; R is for drought-resistant Xinluzhong No. 82; S is for sensitive Kexin No. 1; 1, 2, and 3 are for repeats 1, 2, and 3. This is similar hereafter.

**Table 2 genes-13-01159-t002:** Statistics for the number of differentially expressed genes.

DEG Set	Number of DEGs	Upregulated	Downregulated
XJ0hR1_XJ0hR2_XJ0hR3_vs_XJ0hS1_XJ0hS2_XJ0hS3	2161	1065	1096
XJ0hR1_XJ0hR2_XJ0hR3_vs_XJ12hR1_XJ12hR2_XJ12hR3	16,882	7808	9074
XJ0hR1_XJ0hR2_XJ0hR3_vs_XJ24hR1_XJ24hR2_XJ24hR3	7168	2997	4171
XJ0hS1_XJ0hS2_XJ0hS3_vs_XJ12hS1_XJ12hS2_XJ12hS3	15,626	7498	8128
XJ0hS1_XJ0hS2_XJ0hS3_vs_XJ24hS1_XJ24hS2_XJ24hS3	3070	1224	1846
XJ12hR1_XJ12hR2_XJ12hR3_vs_XJ12hS1_XJ12hS2_XJ12hS3	1614	615	999
XJ12hR1_XJ12hR2_XJ12hR3_vs_XJ24hR1_XJ24hR2_XJ24hR3	15,240	7572	7668
XJ12hS1_XJ12hS2_XJ12hS3_vs_XJ24hS1_XJ24hS2_XJ24hS3	14,396	7159	7237
XJ24hR1_XJ24hR2_XJ24hR3_vs_XJ24hS1_XJ24hS2_XJ24hS3	1563	805	758

Note: DEG Set: the name of the gene set with differential expression; Upregulated: the number of upregulated genes; Downregulated: the number of downregulated genes; and Number of DEGs: the number of differentially expressed genes.

**Table 3 genes-13-01159-t003:** Statistics for the number of lncRNAs that were found to be differently expressed.

DEG Set	Number of DEGs	Upregulated	Downregulated
XJ0hR1_XJ0hR2_XJ0hR3_vs_XJ0hS1_XJ0hS2_XJ0hS3	294	147	147
XJ0hR1_XJ0hR2_XJ0hR3_vs_XJ12hR1_XJ12hR2_XJ12hR3	462	246	216
XJ0hR1_XJ0hR2_XJ0hR3_vs_XJ24hR1_XJ24hR2_XJ24hR3	182	108	74
XJ0hS1_XJ0hS2_XJ0hS3_vs_XJ12hS1_XJ12hS2_XJ12hS3	439	220	219
XJ0hS1_XJ0hS2_XJ0hS3_vs_XJ24hS1_XJ24hS2_XJ24hS3	126	73	53
XJ12hR1_XJ12hR2_XJ12hR3_vs_XJ12hS1_XJ12hS2_XJ12hS3	302	148	154
XJ12hR1_XJ12hR2_XJ12hR3_vs_XJ24hR1_XJ24hR2_XJ24hR3	437	217	220
XJ12hS1_XJ12hS2_XJ12hS3_vs_XJ24hS1_XJ24hS2_XJ24hS3	423	223	200
XJ24hR1_XJ24hR2_XJ24hR3_vs_XJ24hS1_XJ24hS2_XJ24hS3	342	162	180

Note: DEG Set: the name of the gene set that is differentially expressed; Number of DEGs: the number of differentially expressed lncRNAs; Upregulated: the number of lncRNAs that were upregulated; and Downregulated: the number of lncRNAs that were downregulated.

**Table 4 genes-13-01159-t004:** The number of differentially expressed circRNAs and their statistics.

DEG Set	Number of DEGs	Upregulated	Downregulated
XJ0hR1_XJ0hR2_XJ0hR3_vs_XJ0hS1_XJ0hS2_XJ0hS3	8	0	8
XJ0hR1_XJ0hR2_XJ0hR3_vs_XJ12hR1_XJ12hR2_XJ12hR3	11	6	5
XJ0hR1_XJ0hR2_XJ0hR3_vs_XJ24hR1_XJ24hR2_XJ24hR3	1	0	1
XJ0hS1_XJ0hS2_XJ0hS3_vs_XJ12hS1_XJ12hS2_XJ12hS3	2	2	0
XJ0hS1_XJ0hS2_XJ0hS3_vs_XJ24hS1_XJ24hS2_XJ24hS3	6	6	0
XJ12hR1_XJ12hR2_XJ12hR3_vs_XJ12hS1_XJ12hS2_XJ12hS3	11	1	10
XJ12hR1_XJ12hR2_XJ12hR3_vs_XJ24hR1_XJ24hR2_XJ24hR3	5	2	3
XJ12hS1_XJ12hS2_XJ12hS3_vs_XJ24hS1_XJ24hS2_XJ24hS3	0	0	0
XJ24hR1_XJ24hR2_XJ24hR3_vs_XJ24hS1_XJ24hS2_XJ24hS3	11	5	6

Note: DEG Set: the name of the gene set that is differentially expressed; Number of DEGs: the number of differentially expressed circRNAs; Upregulated: the number of circRNAs that were upregulated; and Downregulated: the number of lncRNAs that were downregulated.

**Table 5 genes-13-01159-t005:** Statistical representation of the number of differentially expressed miRNAs.

DEG Set	Number of DEGs	Upregulated	Downregulated
XJ0hR1_XJ0hR2_XJ0hR3_vs_XJ0hS1_XJ0hS2_XJ0hS3	22	8	14
XJ0hR1_XJ0hR2_XJ0hR3_vs_XJ12hR1_XJ12hR2_XJ12hR3	9	3	6
XJ0hR1_XJ0hR2_XJ0hR3_vs_XJ24hR1_XJ24hR2_XJ24hR3	5	3	2
XJ0hS1_XJ0hS2_XJ0hS3_vs_XJ12hS1_XJ12hS2_XJ12hS3	6	3	3
XJ0hS1_XJ0hS2_XJ0hS3_vs_XJ24hS1_XJ24hS2_XJ24hS3	19	13	6
XJ12hR1_XJ12hR2_XJ12hR3_vs_XJ12hS1_XJ12hS2_XJ12hS3	14	7	7
XJ12hR1_XJ12hR2_XJ12hR3_vs_XJ24hR1_XJ24hR2_XJ24hR3	10	7	3
XJ12hS1_XJ12hS2_XJ12hS3_vs_XJ24hS1_XJ24hS2_XJ24hS3	17	7	10
XJ24hR1_XJ24hR2_XJ24hR3_vs_XJ24hS1_XJ24hS2_XJ24hS3	30	15	15

Note: DEG Set: the name of the set of differentially expressed genes; Number of DEGs: the number of differentially expressed miRNAs; Upregulated: the number of upregulated miRNAs; and Downregulated: the number of downregulated miRNAs.

**Table 6 genes-13-01159-t006:** Annotated statistical table of the number of differentially expressed genes.

DEG Set	Total	COG	GO	KEGG	KOG	NR	Pfam	Swiss-Prot	eggNOG
XJ0hR1_XJ0hR2_XJ0hR3_vs_XJ0hS1_XJ0hS2_XJ0hS3	2153	941	1745	782	1101	2153	1832	1776	2056
XJ0hR1_XJ0hR2_XJ0hR3_vs_XJ12hR1_XJ12hR2_XJ12hR3	16,826	7311	13,680	6637	9786	16,822	14,425	13,455	16,415
XJ0hR1_XJ0hR2_XJ0hR3_vs_XJ24hR1_XJ24hR2_XJ24hR3	7143	3241	5971	2867	3686	7143	6196	5830	6950
XJ0hS1_XJ0hS2_XJ0hS3_vs_XJ12hS1_XJ12hS2_XJ12hS3	15,583	6888	12,632	6259	9292	15,579	13,394	12,432	15,198
XJ0hS1_XJ0hS2_XJ0hS3_vs_XJ24hS1_XJ24hS2_XJ24hS3	3054	1394	2527	1280	1518	3054	2638	2541	2968
XJ12hR1_XJ12hR2_XJ12hR3_vs_XJ12hS1_XJ12hS2_XJ12hS3	1602	667	1293	582	910	1602	1358	1296	1509
XJ12hR1_XJ12hR2_XJ12hR3_vs_XJ24hR1_XJ24hR2_XJ24hR3	15,190	6738	12,328	6120	9132	15,188	13,139	12,083	14,848
XJ12hS1_XJ12hS2_XJ12hS3_vs_XJ24hS1_XJ24hS2_XJ24hS3	14,362	6424	11,620	5739	8699	14,357	12,421	11,437	14,033
XJ24hR1_XJ24hR2_XJ24hR3_vs_XJ24hS1_XJ24hS2_XJ24hS3	1557	700	1236	611	889	1557	1328	1255	1487

Note: DEG Set: the name of the gene set that is differentially expressed; Columns 3 to 10 show the number of genes annotated in each functional database.

**Table 7 genes-13-01159-t007:** Annotated statistical table of the number of differentially expressed circRNA source genes.

DEG Set	Total	COG	GO	KEGG	KOG	NR	Pfam	Swiss-Prot	eggNOG
XJ0hR1_XJ0hR2_XJ0hR3_vs_XJ0hS1_XJ0hS2_XJ0hS3	1	1	1	0	1	1	1	1	1
XJ0hR1_XJ0hR2_XJ0hR3_vs_XJ12hR1_XJ12hR2_XJ12hR3	7	5	7	3	7	7	7	7	7
XJ0hR1_XJ0hR2_XJ0hR3_vs_XJ24hR1_XJ24hR2_XJ24hR3	0	0	0	0	0	0	0	0	0
XJ0hS1_XJ0hS2_XJ0hS3_vs_XJ12hS1_XJ12hS2_XJ12hS3	1	1	1	1	1	1	1	1	1
XJ0hS1_XJ0hS2_XJ0hS3_vs_XJ24hS1_XJ24hS2_XJ24hS3	2	2	2	2	2	2	2	2	2
XJ12hR1_XJ12hR2_XJ12hR3_vs_XJ12hS1_XJ12hS2_XJ12hS3	2	2	2	1	2	2	2	2	2
XJ12hR1_XJ12hR2_XJ12hR3_vs_XJ24hR1_XJ24hR2_XJ24hR3	3	2	3	1	3	3	3	3	3
XJ12hS1_XJ12hS2_XJ12hS3_vs_XJ24hS1_XJ24hS2_XJ24hS3	0	0	0	0	0	0	0	0	0
XJ24hR1_XJ24hR2_XJ24hR3_vs_XJ24hS1_XJ24hS2_XJ24hS3	2	2	2	1	2	2	2	2	2

Note: DEG Set: The name of the differentially expressed circRNA set; Total: the total number of annotations; and Columns 3 to 10 indicate the number of circRNA source genes annotated in each functional database (Appendix A).

**Table 8 genes-13-01159-t008:** Annotated statistics on the number of differential miRNA target genes.

DEG Set	Total	COG	GO	KEGG	KOG	NR	Pfam	Swiss-Prot	eggNOG
XJ0hR1_XJ0hR2_XJ0hR3_vs_XJ0hS1_XJ0hS2_XJ0hS3	400	156	299	139	223	400	322	304	377
XJ0hR1_XJ0hR2_XJ0hR3_vs_XJ12hR1_XJ12hR2_XJ12hR3	1009	385	763	377	575	1009	849	793	949
XJ0hR1_XJ0hR2_XJ0hR3_vs_XJ24hR1_XJ24hR2_XJ24hR3	737	285	559	256	419	737	618	574	690
XJ0hS1_XJ0hS2_XJ0hS3_vs_XJ12hS1_XJ12hS2_XJ12hS3	376	127	270	119	229	376	310	309	365
XJ0hS1_XJ0hS2_XJ0hS3_vs_XJ24hS1_XJ24hS2_XJ24hS3	754	269	566	261	439	754	633	589	710
XJ12hR1_XJ12hR2_XJ12hR3_vs_XJ12hS1_XJ12hS2_XJ12hS3	297	109	210	104	181	297	247	226	288
XJ12hR1_XJ12hR2_XJ12hR3_vs_XJ24hR1_XJ24hR2_XJ24hR3	137	46	101	49	75	137	123	118	132
XJ12hS1_XJ12hS2_XJ12hS3_vs_XJ24hS1_XJ24hS2_XJ24hS3	493	185	387	175	321	493	398	404	470
XJ24hR1_XJ24hR2_XJ24hR3_vs_XJ24hS1_XJ24hS2_XJ24hS3	1999	716	1530	702	1175	1999	1645	1547	1900

Note: DEG Set: The name of the differentially expressed miRNA set; Total: the total number of annotations; and Columns 3 to 10 indicate the number of miRNA source genes annotated in each functional database.

**Table 9 genes-13-01159-t009:** Key regulatory genes in the alanine, aspartate, and glutamate metabolism pathway.

Key Gene	Homologous Genes in *Arabidopsis*	Homology Index %
*Gohir.* *A11G156000*	*AT3G22200*	77
*Gohir.* *A11G227533*	*AT3G27740*	79
*Gohir.* *D01G114300*	*AT5G10920*	76
*Gohir.D04G022000*	*AT3G24090*	77
*Gohir.* *D05G088300*	*AT3G47340*	77
*Gohir.* *A07G220600*	*AT5G14760*	76

## Data Availability

The raw sequence data presented in this research have been deposited in the National Center for Biotechnology Information (NCBI) under the accession numbers PRJNA769509 and PRJNA769837, and are publicly accessible at https://www.ncbi.nlm.nih.gov/ (accessed on 1 January 2021).

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
