# Peer review of "Whole Transcriptome Sequencing Reveals Drought Resistance-Related Genes in Upland Cotton"

_genes, 2022, doi:10.3390/genes13071159_

Round 1
Reviewer 1 Report
This study was performed at a good level and it has unique and interesting results. However, the form of presentation of the results is still unacceptable.
The authors modified the manuscript, but not to the full extent as expected. I recommend that authors familiarize with similar articles on transcriptomic analysis of plants and modify the manuscript according to certain standards. Authors should carefully rework the text. Submission of materials and methods is unacceptable.
In the abstract, it is necessary to specify the full Latin name of cotton, so that the reader can further understand what the Gohir genes.
The names of genes and Latin names of plants should be written in italics throughout the text. Latin names of plants are written in full for the first time in the text, then abbreviated (see abstract, lines 24 and 25).
Materials and methods. Lines 90-97 should move it to Chapter 2.1. Plant materials. Delete the chapter "experimental reagents". Lines 85-88 should be deleted, this should be indicated as the methodology is described.
Figure 1 lacks photo control, as it was without drought. And also, photos of plants after 12 and 24 hours of drought.
It is not clear how cDNA was obtained? Which PCR RT program was used. Why was only 1 reference gene used? Usually at least 2 reference genes must be used. How was the statistical calculation carried out? Where is the confidence (p-value) in Figure 6?
p-value italics are required throughout the text (lines 197, 255, 259, and so on).
The abbreviations of the samples must be described for each Figure and Table.
The authors talk about the function of genes with 77% homology with the Arabidopsis gene. In my opinion, 77% is a low threshold to talk about the functions of a gene. It is necessary to translate the nucleotide sequence into an amino acid sequence and already discuss the homology between the amino acid sequences of proteins.
Table 9 lacks the % homology graph.
Also, the authors should carefully review the text for the use of abbreviations. Abbreviations are introduced first, and then they are used in the text (for example, lines 452, 453).
Also, the authors need to issue a manuscript, and in particular a list of references according to the requirements of the journal.
Author Response
Dear Reviewer:
Thank you for your letter and the reviewers’ comments on our manuscript entitled “Whole transcriptome sequencing reveals drought resistance-related genes in upland cotton” (ID: genes-1755694). Those comments are very helpful for revising and improving our paper, as well as the important guiding significance to other research. We have studied the comments carefully and made corrections which we hope meet with approval. The main corrections are in the manuscript and the responds to the reviewers’ comments are as follows.
According to the reviewers’ detailed suggestions, we have made a careful revision on the original manuscript. All revised portions are marked in red in the revised manuscript which we would like to submit for your kind consideration.
Replies to the reviewers’ comments:
Reviewer : In the abstract, it is necessary to specify the full Latin name of cotton, so that the reader can further understand what the Gohir genes.
Response:Thanks for your advice.We have reviewed the manuscript and strengthened the use of language.
The names of genes and Latin names of plants should be written in italics throughout the text. Latin names of plants are written in full for the first time in the text, then abbreviated (see abstract, lines 24 and 25).
Response:Thanks for your advice. It was modified in the manuscript.
Materials and methods. Lines 90-97 should move it to Chapter 2.1. Plant materials. Delete the chapter "experimental reagents". Lines 85-88 should be deleted, this should be indicated as the methodology is described.
Response:Thanks for your advice. It was modified in the manuscript.
Figure 1 lacks photo control, as it was without drought. And also, photos of plants after 12 and 24 hours of drought.
Response: Thanks for your advice. It was modified in the manuscript.
It is not clear how cDNA was obtained? Which PCR RT program was used. Why was only 1 reference gene used? Usually at least 2 reference genes must be used. How was the statistical calculation carried out? Where is the confidence (p-value) in Figure 6.
Response: Thanks for your advice. It was modified in the manuscript.
p-value italics are required throughout the text (lines 197, 255, 259, and so on).
Response: Thanks for your advice. It was modified in the manuscript.
The abbreviations of the samples must be described for each Figure and Table.
Response: Thanks for your advice. It was modified in the manuscript.
The authors talk about the function of genes with 77% homology with the Arabidopsis gene. In my opinion, 77% is a low threshold to talk about the functions of a gene. It is necessary to translate the nucleotide sequence into an amino acid sequence and already discuss the homology between the amino acid sequences of proteins.
Response: Thanks for your advice. It was modified in the manuscript.
表 9 缺少 % 同源图。
Response: Thanks for your advice. It was modified in the manuscript.
Also, the authors should carefully review the text for the use of abbreviations. Abbreviations are introduced first, and then they are used in the text (for example, lines 452, 453).
Response: Thanks for your advice. It was modified in the manuscript.
Also, the authors need to issue a manuscript, and in particular a list of references according to the requirements of the journal.
Response: Thanks for your advice. We have made the necessary modifications according to your comments and suggestions.
Once again, thank you very much for your constructive comments and suggestions which would help us both in English and in depth to improve the quality of the paper.
Kind regards,
Ze-Liang Zhang
邮 箱: zzldeyouxiang@126.com
通讯作者:李雪源
电子邮件地址:xjmh2338@163.com
Reviewer 2 Report
Thank you for responding to the comments.
Line item 93: Drought treatment with 17% PEG, have you done a dose response to come up with that treatment or is there a reference from previously published work?
Line item 255, 259: P-value must be italicized
Regards
Author Response
Dear Reviewer:
Thank you for your letter and the reviewers’ comments on our manuscript entitled “Whole transcriptome sequencing reveals drought resistance-related genes in upland cotton” (ID: genes-1755694). Those comments are very helpful for revising and improving our paper, as well as the important guiding significance to other research. We have studied the comments carefully and made corrections which we hope meet with approval. The main corrections are in the manuscript and the responds to the reviewers’ comments are as follows (the replies are highlighted in blue ).
According to the reviewers’ detailed suggestions, we have made a careful revision on the original manuscript. All revised portions are marked in red in the revised manuscript which we would like to submit for your kind consideration.
Replies to the reviewers’ comments:
Reviewer : Line item 93: Drought treatment with 17% PEG, have you done a dose response to come up with that treatment or is there a reference from previously published work.
Response:Thanks for your advice. We refer to the paper and do relevant experiments.
Reviewer : Line item 255, 259: P-value must be italicized.
Response:Thanks for your advice. It was modified in the manuscript.We have made the necessary modifications according to your comments and suggestions.We hope that the revised manuscript will be approved for publication.
Once again, thank you very much for your constructive comments and suggestions which would help us both in English and in depth to improve the quality of the paper.
Kind regards,
Ze-Liang Zhang
E-mail: zzldeyouxiang@126.com
Corresponding author : Xue-Yuan Li
E-mail address: xjmh2338@163.com
Reviewer 3 Report
The authors of this article have studied the drought induced transcription levels of all the genes of upland cotton. The authors have done a fantastic job by precisely arranging all the data sets. I believe this study will help the related science community for their future research. I do have some minor comments for the authors to consider.
1. I feel, the details of the “Material and methods” section is somehow missing. Please includes details in the section 2.4 and 2.7.
2. The “Conclusion” section is somehow sounds repeat of the discussion section. The authors could consider making it more precise.
Author Response
Dear Reviewer:
Thank you for your letter and the reviewers’ comments on our manuscript entitled “Whole transcriptome sequencing reveals drought resistance-related genes in upland cotton” (ID: genes-1755694). Those comments are very helpful for revising and improving our paper, as well as the important guiding significance to other research. We have studied the comments carefully and made corrections which we hope meet with approval. The main corrections are in the manuscript and the responds to the reviewers’ comments are as follows (the replies are highlighted in blue ).
According to the reviewers’ detailed suggestions, we have made a careful revision on the original manuscript. All revised portions are marked in red in the revised manuscript which we would like to submit for your kind consideration.
Replies to the reviewers’ comments:
Reviewer : I feel, the details of the “Material and methods” section is somehow missing. Please includes details in the section 2.4 and 2.7.
Response:Thanks for your advice. It was modified in the manuscript.
Reviewer : The “Conclusion” section is somehow sounds repeat of the discussion section. The authors could consider making it more precise.
Response:Thanks for your advice. It was modified in the manuscript.We have made the necessary modifications according to your comments and suggestions.We hope that the revised manuscript will be approved for publication.
Once again, thank you very much for your constructive comments and suggestions which would help us both in English and in depth to improve the quality of the paper.
Kind regards,
Ze-Liang Zhang
E-mail: zzldeyouxiang@126.com
Corresponding author : Xue-Yuan Li
E-mail address: xjmh2338@163.com
Round 2
Reviewer 1 Report
The authors have corrected the manuscript according to the recommendations. However, I did not see any chThe authors have corrected the manuscript according to the recommendations. However, I did not see any changes regarding some of the comments"It is not clear how cDNA was obtained? Which PCR RT program was used. Why was only 1 reference gene? How was the statistical calculation carried out? Where is the statistical significance (p-value) in Figure 6?"Also, the materials and methods lack the chapter statistical data analysis.
In addition, authors should carefully revise the text and highlight Latin names in italics (lines 407-411).
The authors should correct the references according to the requirements of the journal. anges regarding some The authors have corrected the manuscript according to the recommendations. However, I did not see any changes regarding some of the comments:
"It is not clear how cDNA was obtained? Which PCR RT program was used. Why was only 1 reference gene? How was the statistical calculation carried out? Where is the statistical significance (p-value) in Figure 6?"
Also, the materials and methods lack the chapter statistical data analysis.
In addition, authors should carefully revise the text and highlight Latin names in italics (lines 407-411).
"It is not clear how cDNA was obtained? Which PCR RT program was used. Why was only 1 reference gene? How was the statistical calculation carried out? Where is the statistical significance (p-value) in Figure 6?"
Also, the materials and methods lack the chapter statistical data analysis.
In addition, authors should carefully revise the text and highlight Latin names in italics (lines 407-411).
The authors should correct the references according to the requirements of the journal.
"It is not clear how cDNA was obtained? Which PCR RT program was used. Why was only 1 reference gene? How was the statistical calculation carried out? Where is the statistical significance (p-value) in Figure 6?"
Also, the materials and methods lack the chapter statistical data analysis.
In addition, authors should carefully revise the text and highlight Latin names in italics (lines 407-411).
The authors should correct the references according to the requirements of the journal.
Author Response
Dear Reviewer:
Thank you for your letter and the reviewers’ comments on our manuscript entitled “Whole transcriptome sequencing reveals drought resistance-related genes in upland cotton” (ID: genes-1692620). Those comments are very helpful for revising and improving our paper, as well as the important guiding significance to other research. We have studied the comments carefully and made corrections which we hope meet with approval. The main corrections are in the manuscript and the responds to the reviewers’ comments are as follows (the replies are highlighted in blue ).
According to the reviewers’ detailed suggestions, we have made a careful revision on the original manuscript. All revised portions are marked in red in the revised manuscript which we would like to submit for your kind consideration.
Replies to the reviewers’ comments:
Reviewer :It is not clear how cDNA was obtained? Which PCR RT program was used. Why was only 1 reference gene? How was the statistical calculation carried out? Where is the statistical significance (p-value) in Figure 6
Response:Thanks for your advice.We adjusted and modified the material method and added the corresponding data in Figure 6.
The materials and methods lack the chapter statistical data analysis.
Response:Thanks for your advice.We have made the necessary modifications according to your comments and suggestions.
In addition, authors should carefully revise the text and highlight Latin names in italics (lines 407-411).
Response:Thanks for your advice.We feel very sorry, we carefully checked many times, and made the corresponding modification
The authors should correct the references according to the requirements of the journal. anges regarding some The authors have corrected the manuscript according to the recommendations.
Response:Thanks for your advice. We have made the necessary modifications according to your comments and suggestions. We hope that the revised manuscript will be approved for publication.
Once again, thank you very much for your constructive comments and suggestions which would help us both in English and in depth to improve the quality of the paper.
Kind regards,
Ze-Liang Zhang
邮 箱: zzldeyouxiang@126.com
通讯作者:李雪源
电子邮件地址:xjmh2338@163.com
This manuscript is a resubmission of an earlier submission. The following is a list of the peer review reports and author responses from that submission.
Round 1
Reviewer 1 Report
Dear Authors,
Overall, nice work. Few minor suggestions:
Some figures are too small to read or understand ( Figure 14 a is impossible)
Entire results were based on hydroponics, it would be nice if you could include a line in the abstract and also the word" hydroponic" in the title.
Line item560: Materials and methods: How did you come up with 17% PEG for drought treatment?
Proof read your manuscript ( For example Line 519 says for hydroponic experiments, I believe you meant from)
Line item 566: Figure 15, not able to distinguish differences
Kind regards!
Author Response
Dear Reviewer:
Thank you for your letter and the reviewers’ comments on our manuscript entitled “Whole transcriptome sequencing reveals drought resistance-related genes in upland cotton” (ID: genes-1692620). Those comments are very helpful for revising and improving our paper, as well as the important guiding significance to other research. We have studied the comments carefully and made corrections which we hope meet with approval. The main corrections are in the manuscript and the responds to the reviewers’ comments are as follows (the replies are highlighted in blue ).
According to the reviewers’ detailed suggestions, we have made a careful revision on the original manuscript. All revised portions are marked in red in the revised manuscript which we would like to submit for your kind consideration.
Replies to the reviewers’ comments:
Reviewer :Some figures are too small to read or understand ( Figure 14 a is impossible)
Response:Thanks for your advice.We are sorry that the picture in the article is not clear due to compression.The clear picture is in the attachment.
Entire results were based on hydroponics, it would be nice if you could include a line in the abstract and also the word" hydroponic" in the title.
Response:Thanks for your advice.We modified it. The word" hydroponic" will in the title and abstract.
Line item560: Materials and methods: How did you come up with 17% PEG for drought treatment?
Response:Thanks for your advice.PEG can simulate different degrees of drought stress through osmotic stress.We proved it through pre-processing experiments 17%PEG on the cotton seedling morphological indicators, root activity, chlorophyll content, antioxidant enzyme and osmotic regulation substances to achieve the effect we need, so use that.
Proof read your manuscript ( For example Line 519 says for hydroponic experiments, I believe you meant from)
Response:Thanks for your advice. We have made the necessary modifications according to your comments and suggestions. We hope that the revised manuscript will be approved for publication.
Line item 566: Figure 15, not able to distinguish differences
Response:Thanks for your advice.We did not pay attention to this detail when editing and made modifications.
Once again, thank you very much for your constructive comments and suggestions which would help us both in English and in depth to improve the quality of the paper.
Kind regards,
Ze-Liang Zhang
E-mail: zzldeyouxiang@126.com
Corresponding author : Xue-Yuan Li
E-mail address: xjmh2338@163.com

Reviewer 2 Report
The present study investigates the transcriptomic analysis of drought-resistant and drought-sensitive cotton (Gossypium hirsutum L.) varieties. The study uses modern research methods, namely transcriptomic analysis and bioinformatic data processing. The authors have received many interesting results. However, the form of presentation of the results was chosen unsuccessfully.
First of all, I strongly advise the authors to have a language check and correct the English of the manuscript since, in many points, it is hard to read the manuscript and there are gross spelling and grammar mistakes that prevent the full appreciation of the scientific study.
Authors should arrange references and a list of references according to the requirements of the journal. The Materials and Methods section needs careful considerations and additional info in all points. This chapter is poorly described, which makes it difficult to adequately assess the work done.
There is a lot of redundant information in the results chapter. I recommend that the authors shorten the results by sending some of the data to supplementary material. The authors should focus on the most striking results that demonstrate the difference in the transcriptome of resistant and sensitive cotton varieties.
Lines 25, 27 Arabidopsis thaliana improved -> Arabidopsis thaliana
Lines 26 give a decryption of GABA on the line 24, and use the abbreviation on the line 27
Lines 37-41 The sentence is too long. Please write it concisely.
Table 1,2,6 Explain the abbreviations given and in the legend to the Tables and further Figures.
Line 172 and further in the text P-value improved -> P-value
Fig.5. What kind of genes were selected, note the statistical differences. Write a more complete legend for this figure. Provide links for the reader to quickly compare the RT PCR verification with transcriptome analysis data. The materials and methods describe in detail how many repetitions were made for each gene, which reference genes were used.
Line 249, 260 blue improved -> green
Fig. 8,9,10 There is not enough explanation for each figure. It is not clear which schedule belongs to which group of cotton varieties.
Line 334 CeRNA improved -> ceRNA
Fig. 11,12 what do these figures mean in the context of your research? What does mean A-I abbreviations?
Lines 386-398 the names of genes should be italicized
Table 9, line 420 Arabidopsis improved -> Arabidopsis
Lines 434,444 Authors should first enter abbreviations, and then use them in the text.
line 464 Arabidopsis thaliana improved -> A. thaliana
line 491 Arabidopsis improved -> Arabidopsis
line 540 genes should be italicized
line 576 Gossypium hirsutum improved -> G. hirsutum
I think that authors should attentively rewrite the text.
Author Response
Dear Reviewer:
Thank you for your letter and the reviewers’ comments on our manuscript entitled “Whole transcriptome sequencing reveals drought resistance-related genes in upland cotton” (ID: genes-1692620). Those comments are very helpful for revising and improving our paper, as well as the important guiding significance to other research. We have studied the comments carefully and made corrections which we hope meet with approval. The main corrections are in the manuscript and the responds to the reviewers’ comments are as follows.
According to the reviewers’ detailed suggestions, we have made a careful revision on the original manuscript. All revised portions are marked in red in the revised manuscript which we would like to submit for your kind consideration.
Replies to the reviewers’ comments:
Reviewer : the authors to have a language check and correct the English of the manuscript since, in many points, it is hard to read the manuscript and there are gross spelling and grammar mistakes that prevent the full appreciation of the scientific study.
Response:Thanks for your advice.We have reviewed the manuscript and strengthened the use of language.
Authors should arrange references and a list of references according to the requirements of the journal. The Materials and Methods section needs careful considerations and additional info in all points. This chapter is poorly described, which makes it difficult to adequately assess the work done.
Response:Thanks for your advice. It was modified in the manuscript.
There is a lot of redundant information in the results chapter. I recommend that the authors shorten the results by sending some of the data to supplementary material. The authors should focus on the most striking results that demonstrate the difference in the transcriptome of resistant and sensitive cotton varieties.
Response:Thanks for your advice. It was modified in the manuscript.
Lines 25, 27 Arabidopsis thaliana improved -> Arabidopsis thaliana
Response: Thanks for your advice. It was modified in the manuscript.
Lines 26 give a decryption of GABA on the line 24, and use the abbreviation on the line 27
Response: Thanks for your advice. It was modified in the manuscript.
Lines 37-41 The sentence is too long. Please write it concisely.
Response: Thanks for your advice. It was modified in the manuscript.
Table 1,2,6 Explain the abbreviations given and in the legend to the Tables and further Figures.
Response: Thanks for your advice. It was modified in the manuscript(Line104-105). XJ represents Xinjiang; 0h, 12h, 24h indicates different treatment times, R for drought resistant Xinluzhong NO.82, S for sensitive Kexin NO.1; 1,2 and 3 for repeats 1,2 and 3.
Line 172 and further in the text P-value improved -> P-value
Response: Thanks for your advice. It was modified in the manuscript.
Fig.5. What kind of genes were selected, note the statistical differences. Write a more complete legend for this figure. Provide links for the reader to quickly compare the RT-PCR verification with transcriptome analysis data. The materials and methods describe in detail how many repetitions were made for each gene, which reference genes were used.
Response: Thanks for your advice. It was modified in the manuscript. Line 215-217, 625-629.
Line 249, 260 blue improved -> green
Response: Thanks for your advice. It was modified in the manuscript.
Fig. 8,9,10 There is not enough explanation for each figure. It is not clear which schedule belongs to which group of cotton varieties.
Response: Thanks for your advice. It was modified in the manuscript.
Line 334 CeRNA improved -> ceRNA
Response: Thanks for your advice. It was modified in the manuscript.
Fig. 11,12 what do these figures mean in the context of your research? What does mean A-I abbreviations?
Response: Thanks for your advice. It was modified in the manuscript.
Lines 386-398 the names of genes should be italicized
Response: Thanks for your advice. It was modified in the manuscript.
Table 9, line 420 Arabidopsis improved -> Arabidopsis
Response: Thanks for your advice. It was modified in the manuscript.
Lines 434,444 Authors should first enter abbreviations, and then use them in the text.
Response: Thanks for your advice. It was modified in the manuscript.
line 464 Arabidopsis thaliana improved -> A. thaliana
Response: Thanks for your advice. It was modified in the manuscript.
line 491 Arabidopsis improved -> Arabidopsis
Response: Thanks for your advice. It was modified in the manuscript.
line 540 genes should be italicized
line 576 Gossypium hirsutum improved -> G. hirsutum
Response: Thanks for your advice. It was modified in the manuscript.
Once again, thank you very much for your constructive comments and suggestions which would help us both in English and in depth to improve the quality of the paper.
Kind regards,
Ze-Liang Zhang
E-mail: zzldeyouxiang@126.com
Corresponding author : Xue-Yuan Li
E-mail address: xjmh2338@163.com

Round 2
Reviewer 2 Report
The authors did not make changes according to my recommendations. In particular:
"Authors should arrange references and a list of references according to the requirements of the journal. The Materials and Methods section needs careful considerations and additional info in all points. This chapter is poorly described, which makes it difficult to adequately assess the work done.
There is a lot of redundant information in the results chapter. I recommend that the authors shorten the results by sending some of the data to supplementary material. The authors should focus on the most striking results that demonstrate the difference in the transcriptome of resistant and sensitive cotton varieties."
Author Response
Dear Reviewer:
Thank you for your letter and the reviewers’ comments on our manuscript entitled “Whole transcriptome sequencing reveals drought resistance-related genes in upland cotton” (ID: genes-1692620). Those comments are very helpful for revising and improving our paper, as well as the important guiding significance to other research. We have studied the comments carefully and made corrections which we hope meet with approval. The main corrections are in the manuscript and the responds to the reviewers’ comments are as follows (the replies are highlighted in blue ).
According to the reviewers’ detailed suggestions, we have made a careful revision on the original manuscript. All revised portions are marked in red in the revised manuscript which we would like to submit for your kind consideration.
Replies to the reviewers’ comments:
Reviewer : Authors should arrange references and a list of references according to the requirements of the journal. The Materials and Methods section needs careful considerations and additional info in all points. This chapter is poorly described, which makes it difficult to adequately assess the work done.
Response:Thanks for your advice. We have made the necessary modifications according to your comments and suggestions.
Reviewer : There is a lot of redundant information in the results chapter. I recommend that the authors shorten the results by sending some of the data to supplementary material. The authors should focus on the most striking results that demonstrate the difference in the transcriptome of resistant and sensitive cotton varieties.
Response:Thanks for your advice. We have made the necessary modifications according to your comments and suggestions.We hope that the revised manuscript will be approved for publication.
Once again, thank you very much for your constructive comments and suggestions which would help us both in English and in depth to improve the quality of the paper.
Kind regards,
Ze-Liang Zhang
E-mail: zzldeyouxiang@126.com
Corresponding author : Xue-Yuan Li
E-mail address: xjmh2338@163.com
